# Racial Difference in the Association of Long-Term Exposure to Fine Particulate Matter (PM_2.5_) and Cardiovascular Disease Mortality among Renal Transplant Recipients

**DOI:** 10.3390/ijerph18084297

**Published:** 2021-04-18

**Authors:** Salem Dehom, Synnove Knutsen, Khaled Bahjri, David Shavlik, Keiji Oda, Hatem Ali, Lance Pompe, Rhonda Spencer-Hwang

**Affiliations:** 1School of Public Health, Loma Linda University, 24951 Circle Drive, Loma Linda, CA 92354, USA; sknutsen@llu.edu (S.K.); kbahjri@llu.edu (K.B.); dshavlik@llu.edu (D.S.); koda@llu.edu (K.O.); lpompe@llu.edu (L.P.); rspencer@llu.edu (R.S.-H.); 2School of Nursing, Loma Linda University, 11262 Campus Street, Loma Linda, CA 92350, USA; 3School of Pharmacy, Loma Linda University, 24745 Stewart Street, Loma Linda, CA 92350, USA; 4Redlands Community Hospital, 305 Terracina Blvd, Redlands, CA 92350, USA; Drhatemali@yahoo.com

**Keywords:** air pollution, cardiovascular disease, mortality, transplantation, racial differences

## Abstract

Ambient air pollutants are known risk factors for cardiovascular disease (CVD) morbidity and mortality with significant racial disparities. However, few studies have explored racial differences among highly susceptible subpopulations, such as renal transplant recipients (RTRs). Despite improvements in quality of life after transplantation, CVD remains the major cause of mortality, especially among Black recipients. This study aimed to evaluate potential racial differences in the association between long-term levels of PM_2.5_ and the risk of all-cause, total CVD, and coronary heart disease (CHD) mortality among RTRs. This retrospective study consists of 93,857 non-smoking adults who received a renal transplant between 2001 and 2015. Time-dependent Cox regression was used to assess the association between annual concentrations of PM_2.5_ and mortality risk. In the multivariable-adjusted models, a 10 μg/m^3^ increase in ambient PM_2.5_ levels found increased risk of all-cause (HR = 3.45, 95% CI: 3.08–3.78), CVD (HR = 2.38, 95% CI: 1.94–2.92), and CHD mortality (HR = 3.10, 95% CI: 1.96–4.90). Black recipients had higher risks of all-cause (HR = 4.09, 95% CI: 3.43–4.88) and CHD mortality (HR = 6.73, 95% CI: 2.96–15.32). High levels of ambient PM_2.5_ were associated with all-cause, CVD, and CHD mortality. The association tended to be higher among Black recipients than non-Blacks.

## 1. Introduction

In recent years, several studies have provided substantial evidence of a positive association between air pollution and cardiovascular disease (CVD), morbidity, and mortality [1,2,3,4,5,6,7,8,9]. In 2015, ambient fine particulate matter (PM_2.5_) was ranked among the top five mortality risk factors worldwide [4]. About 4.2 million deaths and 130.1 million disability-adjusted life-years have been linked to ambient PM_2.5_ levels, causing a 20% increase in death compared with 1990 estimates [4]. Racial disparities in all-cause and cause-specific mortality rates have been documented in the literature [10,11], including CVD mortality [12,13,14]. Yet reasons for these racial differences have not been fully explained [10]. Minorities tend to have greater exposure to much higher levels of ambient air pollutants than non-Hispanic Whites [15,16,17,18], which may put them at higher risk for air pollution-associated disease morbidity and mortality [19], mainly CVD. However, there are few studies which explore the impact of racial differences on these relationships [17,20], especially among highly sensitive populations [21].

Renal transplant recipients (RTRs) are a vulnerable population; however, within this population, there are also racial disparities. In spite of the improvement in overall health, RTRs still have much higher risks of CVD morbidity and mortality when compared to the general population [22,23,24]. Among RTRs, Black recipients tend to have a higher prevalence of CVD risk factors [25] and increased CVD mortality [26]. Between 1990 and 2009, rates of death at 1 year, 3 years, and 5 years after transplant were higher among adult African-American recipients compared with Caucasians [27]. Additionally, after a graft loss, the risk rates of all-cause and CVD-related mortality are significantly higher than among those with a functioning allograft [28]. The incidence rate ratio of End-Stage Renal Disease (ESRD) among Blacks/African Americans has been reported to be 2.9 times higher than among Whites [29]. Moreover, compared with other ethnic groups, Black recipients are less likely to have pre-ESRD care by a nephrologist [29]. Furthermore, once they receive a transplant, Black recipients tend to have the highest rate of CVD mortality after transplant [29]. Despite the improvement in early incidence of rejection, the long-term graft survival is still lower among Black than White transplant recipients [30]. These renal morbidity- and mortality-related trends among Blacks can be partially explained by immunologic risk factors [30], access to health care [29], longer waiting time for a transplant [30,31], lower socioeconomic status [32], lower rate of medication adherence [32], and higher prevalence of comorbidities [29,33]. Although there are many factors that may play a role in driving health disparities among RTRs, one yet to be fully explored is the impact of ambient air pollution.

In previous studies, higher risk rates of mortality associated with ambient air pollutant levels have been observed among sensitive groups, such as renal transplant recipients [8,34,35], smokers [36], females [37,38], African Americans [19] and people with diabetes [39,40]. In addition, long-term exposure to high levels of ambient air pollutants have been found to be associated with increased incidence of type 2 diabetes [41,42,43] and hypertension [44,45], which may put them at higher risk of CVD morbidity and mortality. Furthermore, exposure to high levels of PM_2.5_ has also been linked to worse endothelial function [20], which, together with hypertension and type 2 diabetes, negatively impacts kidney function and survival post-transplant.

To date, very few studies have assessed the potential adverse effects of ambient air pollutants on renal transplant recipients [8,34,35,46], and none have explored the impact of racial differences on these effects. In one study conducted among renal transplant recipients, Spencer-Hwang et al. reported a positive association between ambient O_3_ levels and risk of CHD mortality during the 7 year study period [8]. However, no significant association was found between CHD mortality and ambient PM_10_ levels. Unfortunately, the researchers lacked data on ambient PM_2.5_ levels, which have shown a stronger association with CVD mortality than those reported for PM_10_ in different populations [2,38,47]. In a recent study among kidney transplant recipients in Korea, Kim et al. found a significant positive association between PM_10_ and increased risk of graft failure and all-cause mortality [35]. With limited studies on air pollution among renal transplant recipients in general, and on the effects of fine particulate matter specifically, the purpose of this study was to explore possible racial differences in the association between the long-term effects of ambient fine particulate matter (PM_2.5_) and the risk of all-cause and CVD-related mortality, including CHD, among renal transplant recipients.

## 2. Materials and Methods

Study participants were identified from the U.S. Renal Data System (USRDS), a national data repository containing extensive demographic (including updated residential ZIP codes), diagnostic, hospital, and mortality data for persons living with ESRD [48].

The study population includes kidney transplant recipients, 18 years and older, who had their first kidney transplant procedure between the years of 2001 and 2015, with a minimum of 1 year of graft survival, and who have lived in the contiguous U.S. at the same ZIP code throughout follow-up. Subjects were followed until date of CVD mortality or censoring, which occurred at the time of death due to other causes or at the end of the follow-up period (31 December 2015). Subjects who at the time of transplant had prevalent CVD or were current smokers were excluded. The analysis of smoking status and prevalent CVD cases according to quartiles of annual average of PM_2.5_ (μg/m^3^) are available elsewhere [34]. Thus, the final analytic study population consists of 93,857 non-smoking renal transplant recipients.

### 2.1. Outcome Assessment

Fatal cases of CVD were identified from the USRDS using the ESRD death notification code [49]. Primary cause of death among renal transplant recipients was used to classify study participants into cases and non-cases based on the mortality categories in the ESRD Death Notification form. The following are the assessed fatal CVD outcomes and their definitions.

### 2.2. Total CVD Mortality

Any death in which underlying cause of death was CHD (including myocardial infarction or atherosclerotic heart disease), congestive heart disease, cerebrovascular accident (including intracranial haemorrhage or ischemic brain damage /anoxic encephalopathy), cardiac arrhythmia, or cardiac arrest.

### 2.3. CHD Mortality

Underlying cause of death was myocardial infarction or atherosclerotic heart disease.

### 2.4. CHF Mortality

Underlying cause of death was congestive heart disease.

### 2.5. Pollutant Exposure Assignment

To obtain robust estimates of air pollutants, integrated empirical geographic (IEG) regression models developed elsewhere were used to calculate the annual average concentrations for PM_2.5_, O_3_, and NO_2_ after adjusting for several important geographical factors, including land use and population density [50]. In addition to satellite-derived estimates of air pollution levels, the daily measurements of air pollutants at all Air Quality System (AQS) monitoring sites from U.S. Environmental Protection Agency data repository were used to build the IEG regression models [50]. These estimates can be obtained from the Center for Air Climate and Energy Solutions (https://www.caces.us/data, accessed on 11 July 2019). Further details on the estimation method and model building are explained elsewhere [50]. The annual mean concentrations of ambient PM_2.5_, O_3_, and NO_2_ from 2001 to 2015 were assigned to each individual based on residential ZIP codes using geographic information system (GIS) software. Subject-specific ambient air pollutant annual levels were assigned based on changing attained-age risk sets. These estimates were then merged with USRD data for each subject.

### 2.6. Candidate Confounding Variables

The USRDS database encompasses a wealth of information on several important factors [40] which can be utilized to adjust for potential confounding effects, including demographics, lifestyle factors, medical history, and transplant-related factors. Covariates were added to the models with a priori specification and included age; gender; race; primary cause of ESRD (diabetes, hypertension, primary glomerulonephritis, polycystic kidney disease, other factors); length in years from first ESRD services to first transplant (0–1, 2–5, 6–10, +10 years); donor type (deceased/living); ESRD network categories (low, medium, high standardized transplant ratio); BMI (<18.5, 18.5–<5, 25–<30, 30+); types of anti-rejection medications (cyclosporine (yes/no) and tacrolimus (yes/no)); history of hypertension (yes/no), and history of diabetes (yes/no). Anti-rejection medications were evaluated on an intention-to-treat basis. ESRD regional networks were classified based on their standardized transplant ratio (STR) [51], calculated as the total number of observed first transplants divided by the total number of expected first transplants after adjusting for age and calendar year. More details on the calculation are described elsewhere [51].

In addition to the single-pollutant model, we assessed the associations between PM_2.5_ and mortality outcomes using two-pollutant and three-pollutant models that include the gaseous air pollutants (O_3_ and NO_2_).

### 2.7. Statistical Analysis

Descriptive statistics for demographic and health characteristics in the overall study cohort as well as by race were given as a mean ± standard deviation for continuous variables, and a number with valid percentages for categorical variables. Pearson Chi-square and independent *t*-tests were performed to evaluate the associations between these demographics, health characteristics, and tertiles of annual average PM_2.5_ levels (μg/m^3^) with race after assessing the assumptions of these statistical tests.

Time-dependent Cox-proportional hazard regression models with attained age as the time variable and left truncation by age at transplant were used to estimate the association between PM_2.5_ and risk of all-cause, total CVD, and CHD mortality after adjusting for other important covariates. Ambient air pollutant levels were assigned within Cox regression models as a 1 year average, incrementing yearly for each risk set.

The baseline Cox regression model was developed based on an a priori specification that included PM_2.5_, gender, race, and years since first transplant. Primary cause of ESRD; length in years from first ESRD services and first transplant; donor type; ESRD Network categories; BMI categories, and immunosuppressive medications were added to the final model. Considering the change in ambient levels of PM_2.5_ in the last few decades, PM_2.5_ categories were created by calculating the medians of the 25th, 50th, and 75th percentiles of annual PM_2.5_ concentrations during the 15 year follow-up period. The linearity assumption for the main exposure variable with mortality outcomes was assessed graphically by plotting the estimated coefficients of PM_2.5_ quartiles, and it was fairly met. Regression models with interaction terms between PM_2.5_ and race were evaluated and used to assess the effect of ambient PM_2.5_ on all-cause, total CVD, and CHD morality. Time-dependent variables were included in the final models if the proportionality assumption was not met and the effects were reported at the average age of 60 years as indicated in Tables 2–4.

Sensitivity analyses were also done where we included several socioeconomic variables at the ZIP code level into the basic and full models one at a time using the Centers for Disease Control and Prevention Social Vulnerability Index (2010) Database [52]. These variables included: Proportion of persons below the poverty estimate; proportion of civilians (age +16) unemployed Estimate; per capita income estimate, 2006–2010; proportion of persons with no high school diploma (age 25+). None of these variables changed the main effect of the basic and single-pollutant full models by more than 10% when they were tested individually. Several of these are strongly correlated, and thus measure aspects of the same thing. Thus, as expected, the main effect did not change substantially when adding them all together into the same model (Appendix A).

SAS version 9.4 (SAS Institute, Inc., Cary, NC, USA) was used to perform the main analyses of the study. ArcGIS Desktop Release 10.6 (Esri, Redlands, CA, USA) was used for geocoding air pollutant levels and creating maps.

## 3. Results

### 3.1. Study Population

The study population included 93,695 non-smoking renal transplant recipients (19,769 Black and 73,926 non-Blacks [see footnote Table 1 for details]) from across the continental U.S. who had lived at the same ZIP code during the follow-up period. During a median follow-up of 14.91 years, there were 10,824 total deaths, including 3082 fatal CVD cases, of which 624 were CHD cases. When comparing demographics, health characteristics, and level of ambient PM_2.5_ exposure of the study cohort by race, significant differences in some of these factors were observed. Black transplant recipients were more likely to be obese (BMI ≥ 25); have received a cadaveric donor type; have ESRD attributed to hypertension; have more time elapsed between first ESRD service and first transplant procedure; be registered within ESRD networks with a low or medium transplant ratio and live in the highest PM_2.5_ tertile (Table 1). However, there were no significant differences in age at transplant.

### 3.2. All-Cause Mortality Risk

The strongest association between ambient PM_2.5_ and all-cause mortality was found among Black transplant recipients in the multi-pollutant models, especially when adjusting for NO_2_.

#### 3.2.1. Single-Pollutant Models

The basic model with time interaction showed a strong and significant association between each 10 μg/m^3^ increase in ambient PM_2.5_ and all-cause mortality (HR = 3.30, 95% CI: 3.04–3.58). The risk was somewhat higher for Black recipients (HR =3.74, 95% CI: 3.14–4.44) compared with non-Black recipients (HR = 3.16, 95% CI: 2.87–3.47). These estimates were attenuated in the multivariable-adjusted full model with time interaction (HR = 2.36, 95% CI: 2.17–2.56). When including the interaction term between PM_2.5_ and race, a similar trend was observed (Blacks: [HR = 2.69, 95% CI: 2.28–3.18]; non-Blacks: [HR = 2.27, 95% CI: 2.07–2.48]) (Table 2). The difference by race approached statistical significance (*p*-value of interaction term = 0.06).

#### 3.2.2. Two-Pollutant Models

Compared with the single-pollutant full model, the hazard ratio for all-cause mortality for each 10 μg/m^3^ increase in PM_2.5_, after adjusting for O_3_ and NO_2_ in two separate time interaction-adjusted models, increased by 12.7% after adjusting for O_3_ (HR = 2.66, 95% CI: 2.43–2.91) and by 46.2% in the model with NO_2_ (HR = 3.45, 95% CI: 3.08–3.78) (Table 2). For race-specific risk estimates, adjusting for O_3_, Black recipients had a higher risk associated with each 10 ug/m^3^ increment of PM_2.5_ (HR = 3.13, 95% CI: 2.27–3.99) compared to that found for the same increment among non-Black recipients (HR = 2.54, 95% CI: 2.31–2.80) with a *p*-value of interaction term of 0.03. Similarly, when adjusting for NO_2_, Black recipients had a significantly higher risk of all-cause mortality HR of 4.09 (95% CI: 3.43–4.88) compared with that found for non-Black recipients (HR = 3.26, 95% CI: 2.93–3.63) with a *p*-value of interaction term of 0.02 (Table 2).

#### 3.2.3. Three-Pollutant Models

Despite relationship between the pollutants, the hazard ratio for all-cause mortality for each 10 μg/m^3^ increase in PM_2.5_, after adjusting for O_3_ and NO_2_ in the same time interaction-adjusted model, increased by 60.0% compared with the single-pollutant full model (HR = 3.80, 95% CI: 3.40–4.25) (Table 2). For race-specific risk estimates, adjusting for O_3_ and NO_2_, Black recipients had a higher risk associated with each 10 ug/m^3^ increment of PM_2.5_ (HR = 4.67, 95% CI: 3.89–5.60) compared to that found for the same increment among non-Black recipients (HR = 3.62, 95% CI: 3.22–4.07) with a *p*-value of interaction term of 0.006 (Table 2). In this model, the correlation of PM_2.5_ with NO_2_ and O_3_ are −0.37 and −0.24, respectively.

### 3.3. Total CVD Mortality Risk

Similar to all-cause mortality, strong associations were found between ambient PM_2.5_ and total CVD mortality among Black and non-Black transplant recipients. However, there were no statistically significant racial differences (*p*-values for the interaction terms in single- and multi-pollutant models were greater than 0.05).

#### 3.3.1. Single-Pollutant Models

The basic model showed a strong and significant association between each 10 ug/m^3^ increase in ambient PM_2.5_ and total CVD mortality (HR = 2.77, 95% CI: 2.37–3.22). The risk associated with ambient air pollution was somewhat higher among non-Black recipients (HR = 2.79, 95% CI: 2.33–3.33) than Black transplant recipients (HR = 2.67, 95% CI: 1.85–3.56), but it was not statistically significant (*p*-value for interaction term = 0.89). The estimates were attenuated, but still strong and statistically significant in the multivariable-adjusted full model with time interaction (HR = 1.86, 95% CI: 1.58–2.19) and in the race-specific models (Blacks: [HR = 1.82, 95% CI: 1.33–2.49]; non-Blacks: [HR = 1.87, 95% CI: 1.55–2.25]) (Table 3).

#### 3.3.2. Two-Pollutant Models

Compared with the single-pollutant full model, the hazard ratio for each 10 μg/m^3^ increase in PM_2.5_ for total CVD mortality, after adjusting for O_3_ and NO_2_ in two separate time interaction-adjusted models, increased by 9.1% after adjusting for O_3_ (HR = 2.03, 95% CI: 1.47–2.42) and by 28.0% in the model with NO_2_ (HR = 2.38, 95% CI: 1.94–2.92) (Table 3). For race-specific risk estimates, a similar trend in HRs was observed for Black and non-Black recipients adjusting for O_3_ with HR = 2.03 for both races (Table 3). A stronger effect was observed for both races when adjusting for NO_2_, with HR = 2.38 in Blacks, compared with 2.37 in the non-Black recipients (Table 3).

#### 3.3.3. Three-Pollutant Models

Compared with the single-pollutant full model, the hazard ratio for total CVD mortality for each 10 μg/m^3^ increase in PM_2.5_, after adjusting for O_3_ and NO_2_ in the same time interaction-adjusted model, increased by 37.6% (HR = 2.56, 95% CI: 3.40–4.25) (Table 3). For race-specific risk estimates, adjusting for O_3_ and NO_2_, a similar trend in HRs was observed for Black and non-Black recipients when adjusting for O_3_ and NO_2_ with a *p*-value of interaction term of 0.90 (Table 3). In this model, the correlation of PM_2.5_ with NO_2_ and O_3_ are −0.35 and −0.24, respectively.

### 3.4. CHD Mortality Risk

In contrast to the findings for total CVD mortality, we found significant racial differences in CHD mortality. The strongest associations between ambient PM_2.5_ and total CHD mortality were found among Black transplant recipients in both the single- and multi-pollutant models, with the highest estimate after adjusting for NO_2_.

#### 3.4.1. Single-Pollutant Models

The basic model showed that for each 10 μg/m^3^ increase in ambient PM_2.5_, CHD mortality increased more than threefold (HR = 3.18, 95% CI: 2.28–4.43), but was attenuated in the multivariable-adjusted model (HR = 2.23, 95% CI: 1.55–3.21) (Table 4). Regardless of the decreased strength of association between PM_2.5_ and CHD mortality from the basic model to the full model, the association remained strong and statistically significant. The association with ambient PM_2.5_ among Black transplant recipients was more than fourfold (HR = 4.60, 95% CI: 2.10–10.12), whereas it was less than twofold among the non-Black recipients (HR = 1.88, 95% CI: 1.27–2.79), with a *p*-value for the interaction term of 0.04 (Table 4).

#### 3.4.2. Two-Pollutant Models

Compared with the single-pollutant full model, the strength of association between a 10 μg/m^3^ increase in ambient PM_2.5_ levels and risk of CHD mortality was strengthened after adjusting for NO_2_ (HR = 3.10, 95% CI: 1.96–4.90) (Table 4) but remained unchanged in the two-pollutant model with O_3_ (HR = 2.23, 95% CI: 1.48–3.36). In the race-specific models, the highest associations were observed among Black recipients after adjusting for NO_2_ (HR = 6.73, 95% CI: 2.96–15.32) and O_3_ (HR = 4.64, 95%CI: 2.06–10.46) with *p*-values for the interaction term of 0.03 and 0.04, respectively.

#### 3.4.3. Three-Pollutant Models

When adjusting for O_3_ and NO_2_ in the same multi-pollutant model, the hazard ratio for CHD mortality for each 10 μg/m^3^ increase in PM_2.5_ increased by 38.6% compared with the single-pollutant full model (HR = 3.09, 95% CI: 1.88–5.05) (Table 4). Like the two-pollutant model with NO_2_, the race-specific risk estimates when adjusting for O_3_ and NO_2_, the highest associations were observed among Black recipients after adjusting (HR = 6.73, 95% CI: 2.89–15.70) (Table 4) with a *p*-value of interaction term of 0.03 (Table 4). In this model, the correlation of PM_2.5_ with NO_2_ and O_3_ are −0.27 and −0.24, respectively.

## 4. Discussion

To the best of our knowledge, this is the first study to explore racial differences in the association between ambient PM_2.5_ and risk of premature death due to all-cause, total CVD, and CHD mortality among non-smoking renal transplant recipients. The results from this nationwide retrospective cohort study support the hypothesis that PM_2.5_ is an independent risk factor for all-cause, totalCVD and CHD fatal cases among the renal transplant population. We found detrimental effects on mortality even for PM_2.5_ levels below the current EPA and World Health Organization (WHO) standards, indicating potential reassessment of current guidelines. Moreover, exposure to ambient PM_2.5_ seems to play an important role in furthering racial disparities in all-cause and CVD-related mortality among this vulnerable population.

Our findings suggest that Black renal transplant recipients tend to live in areas with high levels of ambient PM_2.5_, which may be a factor in explaining their increased risk of dying at younger ages compared with other renal transplant recipients. Overall, exposure to high levels of ambient PM_2.5_ was positively associated with increased risk of all-cause, total CVD, and CHD mortality after adjusting for other factors, including race. In addition, we found a statistically significant effect modification by race for all-cause and CHD mortality (Table 4). The strongest associations between air pollution and total-mortality outcomes were observed among Black transplant recipients. For every 10 unit increase in PM_2.5_, the risk of CHD mortality increases by more than fourfold in both single- and two-pollutant models, clearly indicative of a major threat to longevity among this racial group. Among non-Blacks, the corresponding association is only half of that observed among Blacks.

Our findings of racial differences in air pollution exposure are consistent with the results of previous studies among less vulnerable populations in the U.S. [17,20,21,53]. In a recent study, Erqou et al., using data from the Heart Strategies Concentration on Risk Evaluation (HeartSCORE) study, reported a significantly higher exposure to PM_2.5_ and black carbon in African Americans compared to White participants in western Pennsylvania [20]. The team also found that African-American participants were at higher risk of combined-incident CVD events and all-cause mortality than Whites after adjusting for traditional risk factors (HR = 1.45, 95% CI:1.00–2.09). However, this association was somewhat attenuated after adjusting for ambient PM_2.5_ levels (HR = 1.34, 95% CI: 0.91–1.96) [20]. In addition, the researchers found positive associations between median ambient PM_2.5_ and increased risk of elevated blood glucose levels and decreased endothelial function [20].

In a study of Medicare beneficiaries, Di et al. found a significant association between annual exposure to higher ambient levels of PM_2.5_ and increased risk of all-cause mortality (HR = 1.073, 95% CI: 1.071–1.075) [21]. Furthermore, the researchers found that Black Medicare beneficiaries had significantly higher risk of all-cause mortality (HR = 1.208, 95% CI: 1.199–1.217) compared with White beneficiaries (HR = 1.063, 95% CI: 1.060–1.065) [21].

In contrast, Parker et al. found a positive association between annual concentrations of PM_2.5_ and heart disease (HR = 1.18, 95% CI: 1.06–1.31) and all-cause mortality (HR = 1.08, 95%CI: 1.01–1.16) among adults [17]. Nevertheless, no significant difference in the relationship between PM_2.5_ level, heart disease, and all-cause mortality by race/ethnicity was found using a sample from the National Health Interview Survey [17]. However, the hazard ratio for all-cause mortality was slightly higher among non-Hispanic Blacks (HR = 1.14, 95% CI: 0.94–1.37) compared with non-Hispanic White participants (HR = 1.09, 95%CI: 1.01–1.17). The opposite trend in the relationship by race for the association between air pollution and heart disease mortality was observed with non-Hispanic Whites having a higher hazard ratio (HR = 1.20, 95% CI: 1.07–1.35) compared with non-Hispanic Blacks (HR = 1.03, 95% CI: 0.78–1.36) [17]. These weak and non-significant results could be due to the limited adjustment for health-related factors and for the change in air pollution levels over time.

Comparing our results with previous research conducted among various populations, we found stronger effects. This could potentially be explained by the unique characteristics of this vulnerable population. Declining kidney function, prevalence of hypertension and diabetes, and use of immunosuppressive medications are factors that, compared to non-diseased subjects, may make non-smoking renal transplant recipients more sensitive to ambient air pollutants and their effects on cardiovascular disease risk. For Black transplant recipients, additional risk factors, such as immunological causes and higher prevalence of comorbidities, may increase their susceptibility to the adverse effects of ambient air pollution.

Over the years, several biological mechanisms have been proposed to explain the relationship between ambient levels of air pollution and CVD morbidity and mortality. Air pollutants have been linked to increased levels of pulmonary oxidative stress and inflammation, which leads to the release of inflammatory factors and free radicals into the bloodstream, causing cell their ability to pass through the plasma membrane of different body cells and interact with them. These interactions may contribute to thrombosis and atherosclerotic plaque injury [54,55,56]. Vascular inflammation has been associated with the development of atherosclerosis after transplantation, which leads to higher numbers of CVD events [57,58]. The second common explanation is the translocation of air pollution particles into blood circulation and formations that eventually lead to changes in the cardiovascular system [1,55,56]. Additionally, a sudden exposure to high levels of ambient air pollution has been found to be associated with significant changes in the stability of atherosclerotic plaque and the thrombogenicity process, which may trigger CVD events [59]. Also, significant relationships have been reported between PM_2.5_ exposure and the occurrence of ventricular arrhythmias, especially among patients with coronary heart disease [60,61], and with diabetes or impaired glucose tolerance [62]. Moreover, some studies have suggested that Black populations have higher levels of inflammatory biomarkers compared with non-Blacks [63,64], which could explain some of the racial variation in mortality rates.

Finally, in a recent study, Julliard et al. suggested that declining long-term graft survival rates after organ transplantation might be a consequence of shifts in the systemic response from self-tolerance and graft acceptance in the early stages of transplantation to an inflammatory response due to exposure to air pollutants, proteins in the diet, and ultraviolet light exposure [65]. More research is needed to better understand these factors and how they influence biological mechanisms, which may influence racial disparities.

### 4.1. Implications for Health Professionals

The findings of this novel study may increase awareness among health care professionals about the adverse effects of air pollution among renal transplant recipients, and some aspects of racial disparities in post-transplant health outcomes. In addition, our findings should encourage multidisciplinary collaboration to develop strategies to lower the impact of air pollution on health outcomes among this vulnerable group. Furthermore, it may help renal transplant recipients make informed decisions to reduce personal exposure to ambient air pollution, particularly in highly polluted areas, which could lead to fewer hospital visits and lower health care costs.

### 4.2. Study Strengths and Limitations

There are several strengths in this study. All health care professionals in the U.S. are by law required to document all subjects who receive a diagnosis of chronic kidney disease as well as all renal transplant recipients. The use of the USRDS, which includes records for the entire U.S. renal transplant population, makes our study results more generalizable to the renal transplant population as a whole. In addition, this database includes a large number of important variables that allow for the adjustment of several confounding effects. Another strength was the ability to adjust for ambient O_3_ and NO_2_ in the multi-pollutant models, allowing us to see how these pollutants modify the effect of PM_2.5_. Using this database assures generalizability to the U.S. kidney transplant population at large with the same characteristics. Furthermore, ZIP code-specific annual average concentrations of air pollutants were assigned using previously available integrated empirical geographic regression models, adjusting for several geographical factors that had high cross-validation statistics.

However, this study also has some limitations. Only annual ambient pollution concentrations at the ZIP code of residence were available as exposure variables, rather than the levels at the subjects’ residence addresses or short-term levels. Also, this database lacks information on place of work, and thus this was not included in our exposure estimates. However, renal transplant recipients’ residences and workplaces are likely in close proximity, and thus have similar ambient PM_2.5_ concentrations. These limitations are therefore unlikely to greatly influence the mean annual level of ambient air pollutants.

Another limitation is the lack of more specific classification on the death certificate. Moreover, there was no available information on previous smoking status, dietary factors, socioeconomic status, psychosocial factors, and physical activity which are all known to influence both total and CVD mortality. However, we were able to control for BMI, which partially accounts for such lifestyle factors. Additionally, we have done sensitivity analyses on the effect of SES using several socioeconomic variables at the ZIP code level and we did not observe any significant impact on the relationship between PM_2.5_ and mortality risks. These findings are in line with previous studies on transplant recipients published in 2011 [8].

## 5. Conclusions

In this unique study, we explored the possible racial differences in the long-term effects of ambient PM_2.5_ on the risk of total and CVD mortality among renal transplant recipients. Compared to non-Blacks, Black renal transplant recipients had higher risks of both all-cause and CHD mortality associated with PM_2.5_. For total CVD mortality, however, the risk associated with ambient PM_2.5_ is high in both racial groups, and the racial differences are less clear. Further studies are needed to confirm our findings, to assess the potential association among other minorities, and to explore the association between the various PM_2.5_ compositions and health outcomes. Additional studies are also needed to assess the role of smoking history, a strong predictor of fatal CHD, on the racial differences in the association between ambient PM_2.5_ levels and fatal CHD. Ultimately, our findings may contribute to the development of preventive policies and personal strategies to lessen the impact of air pollution on health outcomes in renal transplant recipients. Also, our findings could lead to strategies that will reduce healthcare costs for patients with cardiovascular diseases. Furthermore, we hope these findings will help in addressing some aspects of racial disparity in health outcomes post-transplantation. Finally, managing the modifiable CVD risk factors, including ambient air pollution, may eventually reduce the risk of graft loss and improve the quality of life for this vulnerable population.

## Figures and Tables

**Table 1 ijerph-18-04297-t001:** Demographic and Health Characteristics of Overall Study Cohort According to Race.

Characteristic	Black	Non-Black *	*p*-Value
Age at transplant: [years (mean ± SD)] ^a^	51.3 ± 12.4	51.4 ± 13.2	0.308
Years since first transplant: [years (mean ± SD)] ^a^	6.4 ± 3.6	6.7 ± 3.9	<0.001
Gender: n (%) ^b^			<0.001
Male	11,336 (57.4)	44,785 (60.6)	
Female	8430 (42.6)	29,131 (39.4)	
Donor type: n (%) ^b^			<0.001
Cadaveric	14,820 (75.0)	40,098 (54.3)	
Living	4941 (25.0)	33,732 (45.7)	
BMI categories: n (%) ^b^			<0.001
<18.5	507 (2.7)	2087 (3.0)	
18.5–<25.0	4866 (26.0)	23,806 (33.7)	
25.0–<30.0	5910 (31.6)	23,405 (33.1)	
30.0+	7418 (39.7)	21,395 (30.3)	
Elapsed time between first ESRD service and transplantation: n (%) ^b^			<0.001
0–1 Year	7256 (36.7)	47,092 (63.7)	
2–5 Year	9320 (47.1)	22,275 (30.1)	
6–10 Year	2901 (14.7)	4179 (5.7)	
+10 Year	292 (1.5)	380 (0.5)	
Network categories: n (%) ^b^			<0.001
Low	7697 (38.9)	23,879 (32.3)	
Medium	6971 (35.3)	23,644 (32.0)	
High	5101 (25.8)	26,403 (35.7)	
Primary cause of ESRD: n (%) ^b^			<0.001
Diabetes	5762 (30.5)	20,646 (29.8)	
Hypertension	6787 (35.9)	10,514 (15.2)	
Glomerulonephritis	4443 (23.5)	18,770 (27.1)	
Cystic Kidney	845 (4.5)	10,219 (14.8)	
Other Urologic	109 (0.6)	1601(2.3)	
Other Cause	965 (5.1)	7485 (10.8)	
Tertile of annual average of PM_2.5_ (μg/m^3^) ^b^ [Median (Min-Max)]			<0.001
1st [ 7.9 (2.1–9.3 μg/m^3^)]	3443 (17.7)	27,028 (37.4)	
2nd [ 10.3 (>9.3–11.0 μg/m^3^)]	6928 (35.6)	22,901 (31.7)	
3rd [ 11.9 (>11.0–18.4 μg/m^3^)]	9110 (46.8)	22,274 (30.9)	

^a^ Independent t test; ^b^ Chi-square test; * Includes 67,236 whites, 909 Native Americans, 5232 Asians and 549 other race/ethnicities.

**Table 2 ijerph-18-04297-t002:** Multivariable-Adjusted Hazard Ratios for All-cause Mortality per 10 μg/m^3^ Increment of PM_2.5_: Single-Pollutant and Multi-Pollutant Models by Race.

Model	All-Cause Mortality ^¥^		
	Total Sample ^χ^	Black	Non-Black
	HR (95%CI) (Event/N)	HR (95%CI) (Event/N)	HR (95%CI) (Event/N)
Basic Model *	3.30 (3.04, 3.58) (10,565/92,965)	3.74 (3.14, 4.44) (2520/19,600)	3.16 (2.87, 3.47) (8045/73,365)
Single-Pollutant Full Model **	2.36 (2.17, 2.56) (9744/85,171)	2.69 (2.28, 3.18) (2274/18,023)	2.27 (2.07, 2.48) (7470/67,148)
Full model adjusted for O_3_ **	2.66 (2.43, 2.91) ^€^ (9744/85,171)	3.13 (2.27, 3.99) (2274/18,023)	2.54 (2.31, 2.80) (7470/67,148)
Full model adjusted for NO_2_ **	3.45 (3.08, 3.78) ^€^ (9744/85,171)	4.09 (3.43, 4.88) (2274/18,023)	3.26 (2.93, 3.63) (7470/67,148)
Full model adjusted for NO_2_ & O_3_ **	3.80 (3.40, 4.25) ^€^ (9744/85,171)	4.67 (3.89, 5.60) (2274/18,023)	3.62 (3.22, 4.07) (7470/67,148)

* Adjusted for the following variables: sex, race, and years after transplant; ** Adjusted for all the following variables: sex, years after transplant, primary cause of ESRD, length in years from first ESRD services and first transplant, donor type, ESRD Network categories, BMI categories, and immunosuppressive medications; ^¥^ Effect estimates at attained age of 60 years. Using time-dependent variables. ^€^
*p*-value for interaction term <0.05. **^χ^** Adjusted for race.

**Table 3 ijerph-18-04297-t003:** Multivariable-Adjusted Hazard Ratios for Total CVD Fatal Events per 10 μg/m^3^ Increment of PM_2.5_: Single-Pollutant and Two-Pollutant Models by Race.

Model	Total CVD Mortality ^¥^		
	Total Sample ^χ^	Black	Non-Black
	HR (95% CI) (Event/N)	HR (95% CI) (Event/N)	HR (95% CI) (Event/N)
Basic Model *	2.77 (2.37, 3.22) (3000/92,965)	2.67 (1.85, 3.65) (821/19,600)	2.79 (2.33, 3.33) (2179/73,365)
Single-Pollutant Full Model **	1.86 (1.58, 2.19) (2783/85,171)	1.82 (1.33, 2.49) (743/18,023)	1.87 (1.55, 2.25) (2040/67,148)
Full model adjusted for O_3_ **	2.03 (1.70, 2.42) (2783/85,171)	2.03 (1.47, 2.80) (743/18,023)	2.03 (1.67, 2.47) (2040/67,148)
Full model adjusted for NO_2_ **	2.38 (1.94, 2.92) (2783/85,171)	2.38 (1.70, 3.33) (743/18,023)	2.37 (1.90, 2.96) (2040/67,148)
Full model adjusted for NO_2_ & O_3_ **	2.56 (2.06, 3.19) (2783/85,171)	2.61 (1.85, 3.69) (743/18,023)	2.55 (2.02, 3.22) (2040/67,148)

* Adjusted for the following variables: sex, race, and years after transplant; ** Adjusted for all the following variables: sex, years after transplant, primary cause of ESRD, length in years from first ESRD services and first transplant, donor type, ESRD Network categories, BMI categories, and immunosuppressive medications; ^¥^ Effect estimates at attained age of 60 years. using time-dependent variables. **^χ^** Adjusted for race.

**Table 4 ijerph-18-04297-t004:** Multivariable-Adjusted Hazard Ratios for CHD Fatal Events per 10 μg/m^3^ Increment of PM_2.5_: Single-Pollutant and Two-Pollutant Models by Race.

Model	CHD Mortality ^¥^		
	Total Sample ^χ^	Black	Non-Black
	HR (95% CI) (Event/N)	HR (95% CI) (Event/N)	HR (95% CI) (Event/N)
Basic Model *	3.18 (2.28, 4.43) ^€^ (607/92,965)	7.02 (3.44, 14.31) (130/19,600)	2.59 (1.78, 3.78) (477/73,365)
Single-Pollutant Full Model **	2.23 (1.55, 3.21) ^€^ (564/85,171)	4.60 (2.10, 10.12) (116/18,023)	1.88 (1.27, 2.79) (448/67,148)
Full model adjusted for O_3_ **	2.23 (1.48, 3.36) ^€^ (564/85,171)	4.64 (2.06, 10.46) (116/18,023)	1.90 (1.23, 2.91) (448/67,148)
Full model adjusted for NO_2_ **	3.10 (1.96, 4.90) ^€^ (564/85,171)	6.73 (2.96, 15.32) (116/18,023)	2.62 (1.61, 4.26) (448/67,148)
Full model adjusted for NO_2_ & O_3_ **	3.09 (1.88, 5.05) ^€^ (564/85,171)	6.73 (2.89, 15.70) (116/18,023)	2.63 (1.57, 4.40) (448/67,148)

* Adjusted for the following variables: sex, race, and years after transplant; ** Adjusted for all the following variables: sex, years after transplant, primary cause of ESRD, length in years from first ESRD services and first transplant, donor type, ESRD Network categories, BMI categories, and immunosuppressive medications. ^¥^ Effect estimates at attained age of 60 years. Using time-dependent variables. ^€^
*p*-value for interaction term <0.05. **^χ^** Adjusted for race.

## Data Availability

Renal transplant data can be accessed with a special research proposal and permission from the USRDS (https://www.usrds.org/request.aspx, accessed on 11 April 2021). Air pollution data are available from the Center of Air, Climate, and Energy Solutions (https://www.caces.us/data, accessed on 11 July 2019).

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
