# Peer review of "Racial Difference in the Association of Long-Term Exposure to Fine Particulate Matter (PM2.5) and Cardiovascular Disease Mortality among Renal Transplant Recipients"

_ijerph, 2021, doi:10.3390/ijerph18084297_

Round 1
Reviewer 1 Report
I appreciate the authors' responses to my original comments. However, these responses should have resulted in greater improvements to the manuscript, instead of merely a rebuttal. For instance, if the former smoker data is unavailable, this should be listed as a limitation. Furthermore, the information on why the NO2 and O3 models multi-pollutant models weren't run should be included in the methods section.
Most importantly, I appreciate the authors inclusion of additional sensitivity analyses on key socioeconomic confounders. However, instead of running a series of models adjusting for each factor individually, these factors should all be included in a single model as they each capture different aspects of this important confounder. Additionally, I would expect to see the results from these models in the supplemental materials, given their importance. Merely specifying that they did not significantly change the results is insufficient.
Author Response
Dear Reviewer,
We would like to thank you for taking the time review our manuscript and for the great recommendations and suggestions. Here are the responses for your comments and suggestions.
Point 1:
I appreciate the authors' responses to my original comments. However, these responses should have resulted in greater improvements to the manuscript, instead of merely a rebuttal. For instance, if the former smoker data is unavailable, this should be listed as a limitation.
Response 1:
We have included it on the limitation section (Page 13 - Line 475) and conclusion section (Page 13 – Line 491- 493)
Point 2:
Furthermore, the information on why the NO2 and O3 models multi-pollutant models weren't run should be included in the methods section.
Response 2:
We have included more details under the method section regarding multipollutant models (Page 4 – Lines 150 -152) and in the result section (Pages 7 – 10) as highlighted in the revised manuscript.
Point 2: Most importantly, I appreciate the authors inclusion of additional sensitivity analyses on key socioeconomic confounders. However, instead of running a series of models adjusting for each factor individually, these factors should all be included in a single model as they each capture different aspects of this important confounder. Additionally, I would expect to see the results from these models in the supplemental materials, given their importance. Merely specifying that they did not significantly change the results is insufficient
Response 2:
We have included more details under the method section. Additionally, we have added a supplemental table for the sensitivity analyses for SES status. (Page 4 – lines 179 – 188).
Please let us know if you have any questions or comments.
Best regards,

This manuscript is a resubmission of an earlier submission. The following is a list of the peer review reports and author responses from that submission.
Round 1
Reviewer 1 Report
The study titled, Racial Difference in the Association of Long-Term Exposure to Fine Particulate Matter (PM2.5) and Cardiovascular Disease Mortality among Renal Transplant Recipients, leverages a large database of renal transplant recipients to estimate racial differences in cardiovascular disease-related mortality due to PM2.5. The paper is well-written and provides important knowledge on the racial disparities in air pollution-related mortality among RTRs. However, the authors are missing several key confounders that could greatly impact their results. I also have a few minor comments that require correction.
Introduction.
Line 77: why isn’t reference 34 included here?
Line 102. Should “CVS cases” be “CVD cases”?
Methods:
Line 155: The units for air pollution should be given using the micro symbol not u.
Why were NO2 and O3 not included in the same model?
I see that current smokers were excluded from the study. Was information on former smokers available and how was this data handled? It would be best to adjust for former smoking as a covariate.
Most importantly, while the authors are able to correct for a large number of individual level covariates, they are missing many important confounders that might impact their associations. I would recommend that the authors adjust for neighborhood level confounders and indicators of socioeconomic status, such as poverty level, income, educational attainment, population size, and unemployment. Since they have participant zip codes, they should be able to determine some ecologic socioeconomic indicators. Alternatively, they could stratify their analyses by urban/rural or zip code location. Without adjusting for these important confounders, the authors are likely overestimating the effects of air pollutants on CVD mortality in RTRs.
Results:
It would be nice to know the racial composition of the “Non-Black” participants and to have a sensitivity analysis separating the different races within that category.
The structure of the tables (by model) does not follow the structure of the results (by outcome), which makes it difficult to follow the tables with the text. I would recommend reordering the text or the tables.
Author Response
Point 1: Introduction.
Line 77: why isn’t reference 34 included here?
Response 1: We have added it.
Point 2: Line 102. Should “CVS cases” be “CVD cases”?
Response 2: We have fixed it.
Point 3: Methods:
Line 155: The units for air pollution should be given using the micro symbol not u.
Response 3: We have fixed it.
Point 4: Why were NO2 and O3 not included in the same model?
Response 4: These two pollutants are known to be highly correlated. One of main the sources of Ozone is photochemical reactions between Nitrogen oxides and volatile organic compounds (VOCs)*. Therefore, to avoid multicollinearity issues, we did not run a model with the 3 pollutants.
Point 5: I see that current smokers were excluded from the study. Was information on former smokers available and how was this data handled? It would be best to adjust for former smoking as a covariate.
Response 5: Unfortunately, we do not have information regarding the past smoking history.
Point 6: Most importantly, while the authors are able to correct for a large number of individual level covariates, they are missing many important confounders that might impact their associations. I would recommend that the authors adjust for neighbourhood level confounders and indicators of socioeconomic status, such as poverty level, income, educational attainment, population size, and unemployment. Since they have participant zip codes, they should be able to determine some ecologic socioeconomic indicators.
Response 6: Thank you for the great suggestion. We have added statement regarding the sensitivity analysis (see page 4) where we have adjusted for several of the population SES indicators as advised using CDC Social Vulnerability Index.
Point 7: Results: It would be nice to know the racial composition of the “Non-Black” participants and to have a sensitivity analysis separating the different races within that category.
Response 7: The majority (71.8%) of the non-Blacks are white and only small percentages are Asians (5.6%), Native Americans (0.97%) and other race/ethnicities (0.59%). Due to the very small size and low number of cases among other race/ethnicities beside Black and White, we are not able to run sensitivity for other races. However, we did run an analysis where we limited the analyses to only whites and this the main effect of PM2.5 was very similar to that reported for non-Blacks. Therefore, we decided to include the non-whites among the Non-Black group.
Point 8: Line The structure of the tables (by model) does not follow the structure of the results (by outcome), which makes it difficult to follow the tables with the text. I would recommend reordering the text or the tables.
Response 8: We have modified the tables according to your recommendation. Thus, we have the same structure in the tables as in the text, e.g., each outcome described by basic, single and then two pollutant models.
Reviewer 2 Report
This study is quite interesting and novel which has evaluated potential racial differences in the association between long-term levels of PM2.5 and the risk of cardiovascular disease mortality among renal transplant recipients for the first time. Although, analysis methods used in this study are all relatively traditional, their findings may contribute to the development of personal strategies to reduce the impact of PM2.5 pollution on health outcomes in vulnerable populations.The presentation is well and the written is within a good overall standard of English. However, several parts of this manuscript could be improved.
Major comments:
(1) In section 2.5. Pollutant Exposure Assignment, the air pollution level could be classified according to the WHO Air quality guidelines for particulate matters. It may help the readers to have a better understanding of the exposure level and provide additional information for the relationship between air pollution and health impact among renal transplant recipients.
(2) In section 2.6. Candidate Confounding Variables, more information about the selection of confounding variables should be added. As the author mentioned in the Introduction, several socioeconomic factors may play a role in driving health disparities based on previous studies. However, in the present manuscript, none of socioeconomic factors were included in the set of confounding variables.
(3) In section 4. Discussion, This is a lengthy part containing various analysises. The authors had done a lots of works. However, it is quite difficult for me to identify the most important or valuable point of view of this study. Please refine this part and optimize the structure to show the key findings. I think the main finding of this study should be the “racial difference” , rather than the relationship between air pollution and CVD.
Some Details:
(1) Line 223, the “2” of “NO2” require subscripts for proper rendering.
(2) Line 217, why O3 and NO2 were selected to adjust the model?Are they the major air pollutants in the study area?
(3) Line 367-369, the author has mentioned that “Additionally, a sudden exposure to high levels of ambient air pollution has been found to be associated with significant changes in the stability of atherosclerotic plaque and the thrombogenicity process, which may trigger CVD events.”, thus, it is suggested to add additional adjustment of short-term PM2.5 exposure in the pollutants model of this study if possible.

Author Response
Response to Reviewer 2 Comments
Point 1: (1) In section 2.5. Pollutant Exposure Assignment, the air pollution level could be classified according to the WHO Air quality guidelines for particulate matters. It may help the readers to have a better understanding of the exposure level and provide additional information for the relationship between air pollution and health impact among renal transplant recipients.
Response 1: We have added statement to the article “We found detrimental effects on mortality even for PM2.5 levels below the current EPA and World Health Organization (WHO) standards, indicating potential reassessment of current guidelines.”
|
|
Guidelines |
|
|
|
EPA* |
WHO** |
|
Annual mean of PM2.5 |
≤ 12 µ/m3 |
≤ 10 µ/m3 |
*https://www.epa.gov/pm-pollution/national-ambient-air-quality-standards-naaqs-pm
**https://www.who.int/news-room/fact-sheets/detail/ambient-(outdoor)-air-quality-and-health
Point 2: 2) In section 2.6. Candidate Confounding Variables, more information about the selection of confounding variables should be added. As the author mentioned in the Introduction, several socioeconomic factors may play a role in driving health disparities based on previous studies. However, in the present manuscript, none of socioeconomic factors were included in the set of confounding variables.
This is a good suggestion. However, one of the limitations of this study was that we only had access to the annual ambient pollution concentrations at the ZIP code of residence. Thus, we were not able to single out short-term PM2.5 levels to adjust for these.
However, others have found that there is a relatively high correlation between annual mean concentrations and short-term levels of PM2.5. Therefore, we do not think the relatively strong associations between PM2.5 and various mortality outcomes would be significantly changed even if we adjusted for the short-term effects of this pollutant.
Response 2: Thank you for this comment which is similar to one of the comments from Reviewer 1. We had limited SES factors in the original database, but we have been able to identify and use some population level factors that reflect SES from Centers for Disease Control and Prevention*. They include proportion of persons below the poverty estimate; proportion of civilian (age +16) unemployed estimate; per capita income estimate, 2006-2010; proportion of persons with no high school diploma (age 25+).
We have included each of them, one by one, in sensitivity analyses in the basic and full models and none of them changed the original estimates by more than 8%. These findings are in line with previous transplant recipients study published in 2011**.
*Centers for Disease Control and Prevention/ Agency for Toxic Substances and Disease Registry/ Geospatial Research, A., and Services Program. (2020). CDC Social Vulnerability Index [2010] Database [US]. Retrieved from https://www.atsdr.cdc.gov/placeandhealth/svi/data_documentation_download.html
** Spencer-Hwang, R., et al., Ambient Air Pollutants and Risk of Fatal Coronary Heart Disease Among Kidney Transplant Recipients. American Journal of Kidney Diseases, 2011. 58(4): p. 608-616.
Point 3: In section 4. Discussion, this is a lengthy part containing various analysis. The authors had done a lots of works. However, it is quite difficult for me to identify the most important or valuable point of view of this study. Please refine this part and optimize the structure to show the key findings. I think the main finding of this study should be the “racial difference” , rather than the relationship between air pollution and CVD.
Response 3: We have adjusted the discussion section to focus mainly on the racial differences.
Point 4: (1) Line 223, the “2” of “NO2” require subscripts for proper rendering.
Response 4: We have fixed it.
Point 5: (2) Line 217, why O3 and NO2 were selected to adjust the model?Are they the major air pollutants in the study area?
Response 5: Yes, according, to The United States Environmental Protection Agency (EPA), Particulate Matter, Ozone and Nitrogen Oxides are among the most common outdoor air pollutants in the United States. They are also the ones routinely measured throughout the US and that is the reason these were used in the current analyses.
Point 6: (3) Line 367-369, the author has mentioned that “Additionally, a sudden exposure to high levels of ambient air pollution has been found to be associated with significant changes in the stability of atherosclerotic plaque and the thrombogenicity process, which may trigger CVD events.”, thus, it is suggested to add additional adjustment of short-term PM2.5 exposures in the pollutants model of this study if possible.
Response 6: This is a good suggestion. However, one of the limitations of this study was that we only had access to the annual ambient pollution concentrations at the ZIP code of residence. Thus, we were not able to single out short-term PM2.5 levels to adjust for these.
However, others have found that there is a relatively high correlation between annual mean concentrations and short-term levels of PM2.5. Therefore, we do not think the relatively strong associations between PM2.5 and various mortality outcomes would be significantly changed even if we adjusted for the short-term effects of this pollutant.
